# Pyrrolo[3,2-*b*]pyrrole-1,4-dione (IsoDPP) End Capped with Napthalimide or Phthalimide: Novel Small Molecular Acceptors for Organic Solar Cells

**DOI:** 10.3390/molecules25204700

**Published:** 2020-10-14

**Authors:** Thu Trang Do, Meera Stephen, Khai Leok Chan, Sergei Manzhos, Paul L. Burn, Prashant Sonar

**Affiliations:** 1School of Chemistry and Physics, Queensland University of Technology (QUT), 2 George Street, Brisbane 4001, Australia; tranghcmut@gmail.com; 2Centre for Organic Photonics & Electronics, School of Chemistry and Molecular Biosciences, The University of Queensland, Brisbane 4072, Australia; stephmeera@gmail.com; 3Institute of Materials Research and Engineering (IMRE), 2 Fusionopolis Way, Singapore 138634, Singapore; khaileok@therightu.com; 4Centre Énergie Matériaux Télécommunications, Institut National de la Recherche Scientifique, 1650, Boulevard Lionel-Boulet, Varennes, QC J3X1S2, Canada; sergei.manzhos@emt.inrs.ca; 5Centre for Material Science, Queensland University of Technology (QUT), 2 George Street, Brisbane 4001, Australia

**Keywords:** IsoDPP, napthalimide, phthalimide, non-fullerene, electron acceptors, organic solar cells

## Abstract

We introduce two novel solution-processable electron acceptors based on an isomeric core of the much explored diketopyrrolopyrrole (DPP) moiety, namely pyrrolo[3,2-*b*]pyrrole-1,4-dione (IsoDPP). The newly designed and synthesized compounds, 6,6′-[(1,4-bis{4-decylphenyl}-2,5-dioxo-1,2,4,5-tetrahydropyrrolo[3,2-*b*]pyrrole-3,6-diyl)bis(thiophene-5,2-diyl)]bis[2-(2-butyloctyl)-1*H*-benzo[*de*]isoquinoline-1,3(2*H*)-dione] (NAI-IsoDPP-NAI) and 5,5′-[(1,4-bis{4-decylphenyl}-2,5-dioxo-1,2,4,5-tetrahydropyrrolo[3,2-*b*]pyrrole-3,6-diyl)bis(thiophene-5,2-diyl)]bis[2-(2-butyloctyl)isoindoline-1,3-dione] (PI-IsoDPP-PI) have been synthesized via Suzuki couplings using IsoDPP as a central building block and napthalimide or phthalimide as end-capping groups. The materials both exhibit good solubility in a wide range of organic solvents including chloroform (CF), dichloromethane (DCM), and tetrahydrofuran (THF), and have a high thermal stability. The new materials absorb in the wavelength range of 300–600 nm and both compounds have similar electron affinities, with the electron affinities that are compatible with their use as acceptors in donor-acceptor bulk heterojunction (BHJ) organic solar cells. BHJ devices comprising the NAI-IsoDPP-NAI acceptor with poly(3-*n*-hexylthiophene) (P3HT) as the donor were found to have a better performance than the PI-IsoDPP-PI containing cells, with the best device having a V_OC_ of 0.92 V, a J_SC_ of 1.7 mAcm^−2^, a FF of 63%, and a PCE of 0.97%.

## 1. Introduction

Organic solar cells (OSCs) which are fabricated using solution processing techniques have been considered to be a revolutionary technology for harvesting solar energy and converting it into electrical energy due to their outstanding advantages including low cost, short economic and energy pay-back times, the possibility of using simple manufacturing processes, their light-weight nature, and the capacity for fabrication on flexible substrates for large-area devices [1,2,3]. Among the two main types of OSC devices (bilayer and bulk heterojunction (BHJ)), the BHJ devices, which are comprised of a blend of interpenetrating electron donor (D) and electron acceptor (A) materials, have given the best power conversion efficiencies (PCE), with state-of-the-art PCEs now exceeding 16% for a single junction OSC [4,5]. Semiconducting conjugated polymers and non-polymeric materials are usually used as donors in BHJ devices, with fullerene derivatives, either [6,6]-phenyl-C61 butyric acid methyl ester (PC61BM) or [6,6]-phenyl C71-butyric acid methyl ester (PC71BM), originally employed as the key acceptors. Recently, non-fullerene acceptor-based solar cells have shown PCEs of over 16%, with an increase in efficiency achieved by advances in donor and acceptor molecular design, as well as in device fabrication techniques[5]. Although the role of the electron acceptor is as important as the electron donor for achieving high performance, the development of acceptor materials, as compared with the effort on donor compounds, is still catching up.

In terms of the development of new acceptors for highly efficient OSCs, the replacement of conventional fullerene acceptors with novel non-fullerene ones has gained more attention in recent years because non-fullerene acceptors (NFAs) offer several benefits, including (1) potentially lower synthesis costs, (2) a broader absorption in the visible region, (3) chemical tunability and stability, and (4) facile derivatization and functionalization [6,7,8,9]. There are several ways to tune the properties of non-polymeric NFAs in order to boost the device performance. For example, a desired NFA core chromophore can have its properties engineered by variation of the end-capping units to achieve the desired large absorption coefficient, electron affinity, ionization potential, and electron mobility [6,10,11,12].

The outstanding chemical properties of diketopyrrolopyrrole (DPP) moiety such as strong electron-withdrawing ability, high polarity, high charge carrier mobility, strong intermolecular interactions, and good solubility, have led it to be used for constructing various donor-acceptor-based non-polymeric materials and polymers. These DPP-based conjugated materials have been widely employed for a number of optoelectronic devices such as transistors (p-type, n-type, and ambipolar), solar cells, logic circuits, sensors, memory devices, and photodetectors [13,14,15,16,17,18]. Recently, the OSC devices using DPP-based conjugated polymers acceptors have achieved a PCE of 9.2% [19]. However, polymer semiconductors can be difficult to synthesize reproducibly in pure form as compared with non-polymeric materials. The best performing OSC using a non-polymeric DPP-based acceptor had a PCE of 7.4% [20]. Although the PCE of the DPP-based non-polymeric acceptors have been lower than their polymeric counterparts thus far, the approach has a number of benefits such as good yields, defined molecular structure, and good batch-to-batch reproducibility. The lower PCEs of devices based on the non-polymeric DPP acceptors may arise from a lower electron affinity (higher lowest unoccupied molecular orbital (LUMO) energy) and poor blend film morphology [18]. Thus, there is significant scope for developing new small molecular DPP derivatives that can resolve some of the above mentioned issues.

Although there are many publications on the conventional DPP-based molecular and polymeric systems for various applications, works on its isomer, pyrrolo[3,2-*b*]pyrrole-1,4-dione (IsoDPP), are few [21]. Hence, there is considerable scope for this moiety to be explored as a component of a range of materials and their application in various organic electronic devices. The IsoDPP moiety is a part of the structure of a natural dye that is found in lichens [22]. The difference between the chemical structures of DPP and IsoDPP are in the positions of the carbonyl group and nitrogen atom, which are switched. These differences in structure make IsoDPP an interesting system to compare with the conventional DPP system, in terms of molecular design and synthesis, optoelectronic properties, and solid state packing [23,24]. To date, there are few reports on IsoDPP-based materials designed as NFAs and their use in organic solar cells or other type of devices [21,24,25,26].

Herein, we report the design, synthesis, and characterization of two novel Iso-DPP core-based NFAs. In this design, we used a common IsoDPP core as the acceptor moiety, and 2-(2-butyloctyl)-1*H*-benzo[*de*]isoquinoline-1,3(2*H*)-dione (NAI) and 2-(2-butyloctyl)isoindoline-1,3-dione (PI) as the two end capping moieties. The synthesized materials are named and abbreviated as 6,6′-[(1,4-bis{4-decylphenyl}-2,5-dioxo-1,2,4,5-tetrahydropyrrolo[3,2-*b*]pyrrole-3,6-diyl)bis(thiophene-5,2-diyl)]bis[2-(2-butyloctyl)-1*H*-benzo[*de*]isoquinoline-1,3(2*H*)-dione] (NAI-IsoDPP-NAI) and 5,5′-[(1,4-bis{4-decylphenyl}-2,5-dioxo-1,2,4,5-tetrahydropyrrolo[3,2-*b*]pyrrole-3,6-diyl)bis(thiophene-5,2-diyl)]bis[2-(2-butyloctyl)isoindoline-1,3-dione] (PI-IsoDPP-PI). Both materials were used as non-fullerene acceptors together with poly(3-*n*-hexylthiophene) (P3HT) as the donor in the active layer of organic solar cells (OSCs). Finally, we compare the effect of different end-capping moieties attached to IsoDPP-based acceptors on the performance of the OSCs.

## 2. Results and Discussion

### 2.1. New Non-Fullerene Acceptor Molecular Design and Synthesis

The synthetic routes to the two newly designed electron acceptors NAI-IsoDPP-NAI and PI-IsoDPP-PI are depicted in Scheme 1. In order to improve solution processability, 2-butyloctyl chains were attached to the *N*-atom of the brominated naphthalimide and phthalimide moieties, following reported methods [6,27], to produce alkyl-naphthalimide (1) and alkyl-phthalimide (2). Then, the alkylated derivatives were converted to their respective boronic esters 2-(2-butyloctyl)-6-(4,4,5,5-tetramethyl-1,3,2-dioxaborolan-2-yl)-1*H*-benzo[*de*]isoquinoline-1,3(2*H*)-dione (3) and 2-(2-butyloctyl)-5-(4,4,5,5-tetramethyl-1,3,2-dioxaborolan-2-yl)isoindoline-1,3-dione (4) in quantitative yields under palladium catalyzed conditions using bis(pinacolato)diboron. The dibromo Iso-DPP core was synthesized, following an earlier reported route [28]. Finally, the target compounds, NAI-IsoDPP-NAI (56% yield) and PI-IsoDPP-PI (53% yield), were prepared via classical aqueous Suzuki coupling conditions using the dibromo alkylated IsoDPP (5) and compounds 3 or 4. IsoDPP-NAI and PI-IsoDPP-PI were both found to have good solubility in organic solvents such as chloroform, dichloromethane, toluene, and tetrahydrofuran, thus, making them useful for solution-processed organic electronic devices such as OSCs.

### 2.2. Thermal Properties

Thermogravimetric analysis (TGA) and differential scanning calorimetry (DSC) measurements were performed to evaluate the thermal properties of the electron accepting materials. As shown in Figure 1, the 5% weight loss temperature under nitrogen (decomposition temperature, T_d_) of both compounds was about 450 °C, indicating they have excellent stability towards thermal degradation. The high decomposition temperature for NAI-IsoDPP-NAI and PI-IsoDPP-PI is attributed to their rigid conjugated aromatic backbone. During DSC heating and cooling scans, NAI-IsoDPP-NAI exhibited a sharp melting endothermic transition at 173 °C with broad crystallization peak at 105 °C (Figure 2). Meanwhile, PI-IsoDPP-PI demonstrated clear melting and crystallization peaks at 222 and 200 °C, respectively. Furthermore, PI-IsoDPP-PI had a much sharper exothermic peak as compared with that of the NAI-IsoDPP-NAI. The DSC results clearly reveal that the end capping group has a significant impact on the thermal behavior, as seen in other systems [29]. The PI end capping group enhances the propensity of crystallization as compared with the NAI derivative, which indicates that the materials have different packing behaviors.

### 2.3. Optical Properties

The UV-visible absorption spectra of the acceptors in dilute chloroform solution and corresponding thin films deposited on glass substrates are shown in Figure 3. In solution, both compounds exhibited similar spectra, with maxima at 464 nm for NAI-IsoDPP-NAI and 469 nm for PI-IsoDPP-PI. The molar extinction coefficients of the long wavelength absorption were ≈30,000 M^−1^ cm^−1^ for the two compounds. Moving from solution to solid state, led to a red shift in the absorption onset and maxima of about 50 nm for NAI-IsoDPP-NAI and 17 nm for PI-IsoDPP-PI, which was consistent with planarization of the compounds in solid state. In addition, the UV-visible spectrum of the NAI-IsoDPP-NAI thin film also exhibited a small shoulder on the long wavelength side of the absorption, which could arise from a degree of J-aggregation. The absorption onsets were found to be at ≈610 nm and ≈603 nm, respectively. In this study, the absorption onsets were taken to correspond to the optical gaps ($E_{g}^{opt}$), which were ≈2.0 eV and ≈2.1 eV, respectively. Furthermore, as compared with compounds composed of a DPP core and NAI or PI end-capping groups with *n*-octyl chains [30], the maxima of the long wavelength absorption bands of NAI-IsoDPP-NAI and PI-IsoDPP-PI were blue-shifted absorption band by 125 nm and 149 nm, respectively in solution; and by 91 nm and 139 nm, respectively in thin film state. The wider absorption spectra of NAI-DPP-NAI and PI-DPP-PI resulting in narrow $E_{g}^{opt}$ value which was 1.66 eV for both compounds.

### 2.4. Electrochemical Properties

The relationship among the chemical structure and the electrochemical properties of the newly synthesized materials was studied using cyclic voltammetry (CV) with the compounds dissolved in dichloromethane. The cyclic voltammograms were recorded versus the silver/silver nitrate in acetonitrile reference electrode and the referenced to the ferrocenium/ferrocene (Fc^+^/Fc) redox couple. As shown in Figure 4, NAI-IsoDPP-NAI and PI-IsoDPP-PI exhibited quasi-reversible reduction and oxidation waves, with slight changes occurring between the first and second scans. The oxidation (E_ox_) and reduction (E_red_) onset values were obtained from the voltammagrams and used to estimate the electron affinity and ionization potentials of the materials based on the IP of ferrocene being 4.8 eV [31] below the vacuum level, that is, IP = (E_onset(ox)_ + 4.8) eV and EA = (E_onset(red)_ + 4.8) eV. The obtained IP and EA levels for NAI-IsoDPP-NAI and PI-IsoDPP-PI were identical at ≈5.8 eV and ≈3.8 eV, respectively. The high IP values of both materials were due to the strong electron withdrawing nature of the electron acceptor end groups. The calculated EAs indicate that the materials could be used as electron acceptors for a range of commonly used donors. In comparison with the IP/EA values of NAI-DPP-NAI (*n*-octyl) and PI-DPP-PI (*n*-octyl), which were 5.20/3.58 eV and 5.13/3.54 eV [30], respectively, our newly designed compounds showed larger IP and EA values, resulting from changing the core from DPP to IsoDPP. The optical (absorbance maxima and optical gap) and electrochemical (IPs and EAs, electrochemical gap) properties of the molecules are summarized in Table 1. The energy levels of the acceptor materials were compared with those of the classical donor polymer, P3HT, with an energy level diagram of the materials and the OSC electrodes shown in Figure 5.

### 2.5. Ab Initio Calculations

In order to further understand the distribution of the frontier molecular orbitals of the materials, density functional theory [33,34] calculations using the B3LYP [35] and Lanl2dz basis set were undertaken. A polarized continuum model (PCM) [36] was also used to mimic the chloroform solvent. The calculations were performed in Gaussian 09 [37]. To simplify the calculations, the 2-butyloctyl and decyl chains were replaced by shorter ethyl groups, as they had negligible effect on electronic properties at the molecular level in vacuum or solution. The frontier molecular orbital distributions of NAI-IsoDPP-NAI and PI-IsoDPP-PI are shown in Figure 6. The HOMO and LUMO of both compounds were found to be delocalized along the entire conjugated backbone. The HOMO and LUMO energy levels from the calculations for NAI-IsoDPP-NAI were −5.93 and −3.53 eV, respectively. Meanwhile, those of PI-IsoDPP-PI were calculated to be −5.96 eV for the HOMO and −3.50 eV for the LUMO. The fact that the HOMO and LUMO energies were similar for both compounds was consistent with the electrochemical experiments and the fact that the solution UV-visible absorption measurements (Figure 3) were essentially the same. Finally, the dihedral angles between end-capping groups and central building block IsoDPP were found to be relatively small at 21.9° for NAI-IsoDPP-NAI and 17.8° for PI-IsoDPP-PI.

### 2.6. Photovoltaic Properties

To evaluate the photovoltaic performance of the new electron acceptors, bulk heterojunction organic solar cells (OSCs) devices were fabricated with regio-regular P3HT as the donor. The devices had an architecture of ITO/PEDOT: PSS/P3HT: Acceptor/Ca/Al as shown in Figure 7a. The OSCs were fabricated with varying acceptor loading, with the results summarized in Table 2. The power conversion efficiency (PCE) of the NAI-IsoDPP-NAI-based OSCs were found to improve with increasing acceptor loading, while that of PI-IsoDPP-PI-based OSCs were found to reduce with increasing acceptor loading. An acceptor loading of 80% by weight (donor/acceptor ratio of 1:4 by weight) yielded the highest PCE for the NAI-IsoDPP-NAI-based OSCs, with champion device having the J_sc_ = 1.7 mA/cm^2^, V_oc_ = 0.92 V, FF = 63%, and PCE = 0.97%. In contrast, for the PI-IsoDPP-PI-based OSC, an acceptor loading of 50 wt% (donor/acceptor ratio of 1:1 by weight) yielded the highest PCE with the champion device showing a J_sc_ = 1.5 mA/cm^2^, V_oc_ = 0.91 V, FF = 39%, and PCE = 0.52%. Further increase in PI-IsoDPP-PI loading led to a reduction in extracted photocurrent resulting in reduced PCEs. Such a contrast in trends of PCE with acceptor loading could be brought about by dissimilar donor/acceptor blend morphologies stemming from the difference in the propensity of the two acceptor materials to crystallize. From the DSC measurements, it can be seen that NAI-IsoDPP-NAI crystallizes more slowly and this could possibly facilitate better intermixing of the donor and acceptor phases even at the higher loading concentrations. In contrast, the DSC trace for PI-IsoDPP-PI indicates that it has a strong drive to crystallize, and hence at higher loadings would be expected to give rise to larger pure acceptor domains. Such phase separation would lead to the observed decrease in J_sc_ and fill factor. It is notable that OSCs made with NAI-IsoDPP-NAI and PI-IsoDPP-PI acceptors show a V_oc_ of ca. 350 mV higher with respect to those with PCBM acceptors [38]. Such a relative increase in V_oc_ could be due to differences in electron affinities (PCBM has been reported to different EAs dependent on the measurement method [39]) or other factors known to directly or indirectly influence the open circuit voltage in organic solar cells, such as differences in the density of states, energetic disorder, charge transfer states, donor–acceptor interface, and microstructure [40]. The current density–voltage plots and corresponding external quantum efficiency spectra for the best performing devices are shown in Figure 7b,c, respectively. The overlap of the P3HT [41] and NAI-IsoDPP-NAI and PI-IsoDPP-PI acceptor absorption makes it hard to disentangle the contribution of current originating from photoinduced hole and electron transfer from the EQE spectrum.

It is well known that the bulk and interfacial morphology of the active layer plays an important role in influencing device performance. Therefore, we studied the surface morphology of the blend films of P3HT/NAI-IsoDPP-NAI and P3HT/PI-IsoDPP-PI with ratios of 1:4 and 1:1, respectively (the combination that gave the best performance in each case), using atomic force microscopy (AFM) (see Appendix A). The AFM images of the P3HT/NAI-IsoDPP-NAI blended film had clearer phase separation on the film surface, with nanoscale domains observable. Furthermore, the root mean square (RMS) surface roughness of the NAI-IsoDPP-NAI and PI-IsoDPP-PI blend films were 2.78 and 1.95 nm, respectively. The degree of phase separation observed for the P3HT/NAI-IsoDPP-NAI blend film could provide an explanation for the improved photovoltaic performance of the resulting devices, as reported in previous studies [42].

## 3. Conclusions

In conclusion, two new acceptors based on an isoDPP central building block and napthalimide or phthalimide end-capping groups, namely NAI-IsoDPP-NAI and PI-IsoDPP-PI, were synthesized via the classical Suzuki coupling reaction and explored for their potential for use as non-fullerene acceptors for organic solar cells. Both materials displayed good solubility in organic solvents, high thermal stability, and thin film absorption from 300 to 600 nm. The different terminal groups did not change the ionization potentials and electron affinities, which were similar for both compounds. The electron affinities of the materials were suitable to enable photoinduced electron transfer from P3HT and the open circuit voltage was found to be higher than devices using PC61BM as the acceptor due to their smaller electron affinity. Polymer solar cells, based on the configuration of ITO/PEDOT:PSS/P3HT:Acceptor/Ca/Al, were fabricated to investigate the potential application of new materials in solution processed OSCs. The device with NAI-IsoDPP-NAI as the acceptor achieved the highest performance with a V_OC_ of 0.92 V, J_SC_ of 1.7 mAcm^−2^, FF of 63%, and PCE of 0.97% for the champion device. The lower efficiency of the PI-IsoDPP-PI-based cells was attributed to increased phase separation in the blended film due to its greater propensity to crystallize. The relatively high Voc’s are promising, but the low FF and J_sc_ indicate impeded charge generation and/or extraction. Nevertheless, the acceptors add valuable knowledge regarding the design requirements of non-fullerene acceptors and are an important early contribution in the development of semiconducting isoDPP-based materials.

## 4. Experimental Section

### 4.1. Materials and Instruments

All starting materials were purchased from commercial sources as analytical reagents and used directly without any further purification. The 2,7-bis-(4,4,5,5-tetramethyl-[1,3,2] ioxaborolane-2-yl)-fluoren-9-one was prepared according to the literature [43]. H-NMR and C-NMR spectra were obtained with a Varian-400 or Bruker-600 spectrometer (Bruker, Billerica, MA, USA) in deuterated chloroform referenced to 7.26 ppm for H and 77.0 ppm for C. High resolution mass spectra were recorded on an Orbitrap Elite mass spectrometer (Thermo Fisher Scientific, Waltham, MA, USA) equipped with an atmospheric pressure chemical ionization (APCI) source, operating in the positive ion mode at a resolution of 120,000 (at *m*/*z* 400). Thermal analysis was performed using a Pegasus Q500TGA thermogravimetric analyzer under nitrogen atmosphere at a heating rate of 10 °C/min. Differential scanning calorimetry (DSC) was conducted under nitrogen using a Chimaera Q100 DSC. The sample was heated at 10 °C/min from 30 °C to 300 °C. Absorption spectra were recorded on a Carry50 UV-Vis spectrophotometer (Varian, Inc., Palo Alto, CA, USA). Cyclic voltammetry was carried out using tetrabutylammonium hexafluorophosphate (0.1 M) as supporting electrolyte in freshly distilled dichloromethane (from calcium hydride under argon) with a three-electrode cell comprised of a glassy carbon working electrode, a platinum-wire counter electrode, and a silver/silver nitrate in acetonitrile as the reference electrode.

### 4.2. Device Fabrication and Testing

The solar cells were fabricated on pre-patterned indium tin oxide (ITO) glass substrates (15 Ω sq^−1^: Xinyan) in a class 1000 clean room. The substrates were cleaned in a detergent bath (Alconox) at 80 °C for 10 min, followed by sonication in sequence with Alconox, deionized water, acetone, and 2-propanol for 10 min each. The cleaned substrates were dried with nitrogen before spin-coating the subsequent layers. A 30 ± 5 nm thick PEDOT:PSS (Heraeus Clevios P VP AI 4083) layer was deposited by spin-coating at 5000 rpm for 30 s and annealed at 130 °C for 10 min. After being allowed to cool, the substrates were transferred into a nitrogen filled glove box for the rest of device fabrication (O_2_ < 5 ppm, H_2_O < 1 ppm). The 32 mg/mL solutions of P3HT (American Dye Source, Baie d’Urfe, PQ, Canada), NAI-IsoDPP-NAI, and PI-IsoDPP-PI were made up in anhydrous 1,2-dichlorobenzene and stirred at 70 °C, for 1 h, followed by stirring at room temperature overnight. Then, the P3HT/PI-IsoDPP-PI and P3HT/NAI-IsoDPP-NAI solutions were formed by mixing the P3HT and PI-IsoDPP-PI or NAI-IsoDPP-NAI solutions in different ratios (1:1 (*v*/*v*), 1:3 (*v*/*v*), and 1:4 (*v*/*v*)) before being stirred, for 1 h at room temperature. Then, the active layer solutions were filtered through a 0.2 μm PTFE filter and spin-coated onto the PEDOT/PSS film at 1250 rpm for 30 s followed by annealing at 110 °C for 5 min. The substrates where, then, loaded into the evaporation chamber to deposit 20 nm Ca followed by 100 nm Al at a pressure of 1 × 10^−6^ mbar to provide active areas of 0.2 cm^2^.

The current density–voltage characteristics for unencapsulated devices were obtained using a Keithley 2400 source and measurement unit under 1 sun illumination (AM 1.5 G, 100 mW/cm^2^). The solar simulator (Sun 2000, class AAB Abet Technologies, Milford, CT, USA) was calibrated with a National Renewable Energy Laboratory (NREL) certified standard 2 × 2 cm filter-less silicon reference cell. External quantum efficiency (EQE) spectra of the devices were recorded with a PV Measurements Inc. QEX7 system without white light bias, which was calibrated with an NREL certified photodiode without light bias. The performance of the devices was measured in a glovebox with O_2_ < 3 ppm and H_2_O < 1 ppm.

The AFM samples were prepared by spin coating the blend solution on the PEDOT/PSS layer. Tapping mode AFM (NanoScope III, Dimension, Digital Instrument Inc., Tonawanda, NY, USA) was carried out with commercially available tapping mode tips.

### 4.3. Synthesis

The synthetic route for the preparation of dibromo IsoDPP is described in Supporting Information (Scheme S1) and follows reported procedures, and hence only summarized here [25,28]. Firstly, 4-decylaniline (**1A**) and oxalylchloride were reacted in presence of phosphorous pentachloride (PCl_5_) in toluene to provide the *N*,*N*′-bis(4-alkylphenyl)ethanediimidoyl dichloride intermediate (**2A**). Afterwards, ethyl 2-(thiophen-2-yl)acetate (**3A**) was coupled with compound **2A** using 2 M lithium di-*iso*-propylamide (LDA) to form 1,4-bis(4-decylphenyl)-3,6-di(thiophen-2-yl)pyrrolo[3,2-*b*]pyrrole-2,5(1*H*,4*H*)-dione (**4A**). Then, compound **4A** was brominated with *N*-bromosuccinimide (NBS) in chloroform and washing with hot solution of methanol and precipitated to yield a pure yellow-orange solid of 3,6-bis(5-bromothiophen-2-yl)-1,4-bis(4-decylphenyl)pyrrolo[3,2-*b*]pyrrole-2,5(1*H*,4*H*)-dione (**5**) in a good yield (97%) for the last step.

#### 4.3.1. Synthesis of Compound **3**

Anhydrous 1,4-dioxane (20 mL) was added to a mixture of 6-bromo-2-(2-butyloctyl)-1*H*-benzo[de]isoquinoline-1,3(2*H*)-dione 1 (0.3 g, 0.7 mmol), diborane pinacol ester (0.3 g, 1.0 mmol), KOAc (0.3 g, 2.6 mmol), and Pd(dppf)Cl_2_ (35 mg, 0.05 mmol), and the reaction mixture was stirred at 100 °C for 24 h. The reaction was cooled to room temperature, and then filtered to remove the potassium acetate. The solution was concentrated by vacuum evaporation and used directly for the subsequent reaction due to the instability of **3** on silica gel (≈350 mg, ≈quantitative).

#### 4.3.2. Synthesis of Compound **4**

Anhydrous 1,4-dioxane (20 mL) was added to a mixture of 5-bromo-2-(2-butyloctyl)isoindoline-1,3-dione 2 (0.3 g, 0.8 mmol), diborane pinacol ester (0.3 g, 1.1 mmol), KOAc (0.4 g, 3.4 mmol), and Pd(dppf)Cl_2_ (50 mg, 0.06 mmol), and the reaction mixture was stirred at 100 °C for 24 h. The reaction was cooled to room temperature, and then filtered to remove the potassium acetate. The solution was concentrated by vacuum evaporation and used directly for the subsequent reaction due to the instability of **4** on silica gel (≈ 340 mg, ≈ quantitative).

#### 4.3.3. Synthesis of Compound **5**

In the round-bottom flask, 1,4-bisdecyl-3,6-dithiophen-2-yl-diketopyrrolo[3,2-*b*]pyrrole (3.57 g, 4.87 mmol), CHCl_3_ (100 mL), and NBS (1.86 g, 10.52 mmol) were added in the dark. The mixture was stirred at room temperature for 24 h and the solution turned a deep red color. The reaction mixture solution filtered over filter paper to get rid of minute particles and the filtrate was allowed to flow into a swirling solution of methanol (400 mL). The orange precipitate was filtered and washed with hot solution of methanol (400 mL). Around 4.16 g, (yield 97%) of pure dirbromo IsoDPP was collected after drying. ^1^H-NMR (300 MHz, CD_2_Cl_2_, ppm) δ 7.30–7.27 (d, *J* = 8.6 Hz, 4H), 7.22–7.19 (d, *J* = 8.6 Hz, 4H), 6.70–7.68 (d, *J* = 4.1 Hz, 2H), 5.97–5.96 (d, *J* = 4.1 Hz, 2H), 2.72–2.67 (t, *J* = 7.4 Hz, 4H), 1.66 (m, 4H), 1.29 (m, 28H), 0.91–0.86 (t, *J* = 6.6 Hz, 6H). ^13^C-NMR (75 MHz, CD_2_Cl_2_, ppm) δ 170.1, 144.3, 142.7, 131.4, 131.2, 129.8, 129.5, 129.3, 127.4, 115.9, 100.0, 35.6, 32.0, 31.6, 29.8, 29.7, 29.6, 29.4, 29.2, 22.8, 14.0.

#### 4.3.4. Synthesis of NAI-IsoDPP-NAI

Anhydrous 1,4-dioxane (20 mL) was added to a mixture of 6-bromo-2-(2-butyloctyl)-1*H*-benzo[*de*]isoquinoline-1,3(2*H*)-dione **1** (0.3 g, 0.7 mmol), diborane pinacol ester (0.3 g, 1.0 mmol), KOAc (0.3 g, 2.6 mmol), and Pd(dppf)Cl_2_ (35 mg, 0.05 mmol), and the reaction mixture was stirred at 100 °C for 24 h. The reaction was cooled to room temperature, and then filtered to remove the potassium acetate. The solution was concentrated by vacuum evaporation and used directly for the subsequent reaction due to the instability of **3** on silica gel (≈350 mg, ≈quantitative). A solution of 3,6-bis(5-bromothiophen-2-yl)-1,4-bis(4-decylphenyl)pyrrolo[3,2-*b*]pyrrole-2,5(1*H*,4*H*)-dione **5** (120 mg, 0.14 mmol), **3** (≈350 mg), aqueous K_2_CO_3_ (2 M, 10 mL), and deoxygenated toluene (20 mL) was purged with argon for 30 min. Then, tetrakis(triphenylphosphine)palladium(0) (18 mg 0.015 mmol) was added and further deoxygenated by placement under vacuum and back filling with argon three times before being heated at 110 °C under argon for 2 d. The reaction mixture was allowed to cool to room temperature and poured into water (≈50 mL). The mixture was extracted with chloroform (2 × 200 mL). The chloroform extracts were combined, washed with brine (100 mL), dried over anhydrous Na_2_SO_4_, and filtered. The filtrate was collected, and the solvent was completely removed to give a black residue. The black residue was purified using column chromatography (SiO_2_, hexane/CF/EA = 9:1:0.5) to give a reddish powder that was, then, washed with hot acetone to afford the title compound (110 mg, 56%). ^1^H-NMR (600 MHz, CDCl_3_, ppm): δ 8.65–8.63 (d, *J* = 7.1 Hz, 2H), 8.57–8.53 (m, 4H), 7.75–7.73 (t, *J* = 7.4 Hz, 2H), 7.69–7.68 (d, *J* = 7.6 Hz, 2H), 7.35–7.31 (m, 8H), 7.04 (d, *J* = 3.9 Hz, 2H), 6.64 (d, *J* = 3.8 Hz, 2H), 4.13–4.12 (d, *J* = 7.3 Hz, 4H), 2.70–2.68 (t, *J* = 7.6 Hz, 4H), 2.00 (m, 2H), 1.62 (m, 4H), 1.20 (m, 60H), 0.88–0.83 (m, 18H). ^13^C-NMR (150 MHz, CDCl_3_, ppm): δ 170.1, 164.5, 164.2, 144.0, 143.1, 142.5, 138.3, 132.0, 131.9, 131.4, 131.2, 130.6, 130.4, 129.6, 129.3, 128.9, 128.5, 128.4, 127.4, 127.2, 123.1, 122.1, 100.2, 44.6, 36.6, 35.6, 31.9, 31.8, 31.7, 31.5, 31.4, 29.7, 29.6, 29.6, 29.5, 29.3, 29.2, 28.7, 26.5, 23.1, 22.7, 22.6, 14.1, 14.1. λ_max_ (CH_3_Cl/nm): 464 (log ε/dm^3^/mol/cm 4.48). HRMS (EI, *m*/*z*) anal. calcd. for C_94_H_114_N_4_O_6_S_2_+ (M^+^): 1459.8309 (100%); found: 1459.8253 (100%).

#### 4.3.5. Synthesis of PI-IsoDPP-PI

Anhydrous 1,4-dioxane (20 mL) was added to a mixture of 5-bromo-2-(2-butyloctyl)isoindoline-1,3-dione **2** (0.3 g, 0.8 mmol), diborane pinacol ester (0.3 g, 1.1 mmol), KOAc (0.4 g, 3.4 mmol), and Pd(dppf)Cl_2_ (50 mg, 0.06 mmol), and the reaction mixture was stirred at 100 °C for 24 h. The reaction was cooled to room temperature, and then filtered to remove the potassium acetate. The solution was concentrated by vacuum evaporation and used directly for the subsequent reaction due to the instability of **4** on silica gel (≈340 mg, ≈quantitative). A solution of 3,6-bis(5-bromothiophen-2-yl)-1,4-bis(4-decylphenyl)pyrrolo[3,2-*b*]pyrrole-2,5(1*H*,4*H*)-dione **5** (150 mg, 0.17 mmol), **4** (≈ 340 mg), and aqueous K_2_CO_3_ solution (2 M, 10 mL) and deoxygenated toluene (20 mL) was purged with argon for 30 min. Then, tetrakis(triphenylphosphine)palladium (18 mg 0.015 mmol) was added and further deoxygenated by placement under vacuum and back filling with argon three times before being heated at 110 °C under argon for 2 d. The reaction mixture was allowed to cool to room temperature and poured into water (≈50 mL). The mixture was extracted with chloroform (2 × 200 mL). The chloroform extracts were combined, washed with brine (100 mL), dried over anhydrous Na_2_SO_4_, and filtered. The filtrate was collected, and the solvent was completely removed to give a black residue. The black residue was purified using column chromatography (SiO_2_, hexane/CF/EA = 9:1:0.5) to give a dark reddish powder which was, then, washed with hot acetone to afford the title compound (120 mg, 53%). ^1^H-NMR (600 MHz, CDCl_3_, ppm): δ 7.92 (m, 2H), 7.78–7.75 (m, 4H), 7.31–7.26 (m, 8H), 7.13–7.12 (d, *J* = 4.0 Hz, 2H), 6.45 (d, *J* = 4.0 Hz, 2H), 3.57–3.56 (d, *J* = 7.3 Hz, 4H), 2.72–2.70 (t, *J* = 7.6 Hz, 4H), 1.87 (m, 2H), 1.66 (m, 4H), 1.27 (m, 60H), 0.89–0.85 (m, 18H). ^13^C-NMR (150 MHz, CDCl_3_, ppm): δ 170.0, 168.2, 168.2, 144.2, 144.1, 143.0, 139.7, 133.2, 131.5, 131.1, 131.0, 130.3, 130.2, 129.3, 127.4, 124.9, 123.8, 119.7, 100.2, 42.4, 37.0, 35.6, 31.9, 31.8, 31.5, 31.4, 31.1, 29.7, 29.6, 29.5, 29.4, 29.2, 28.5, 26.3, 23.0, 22.7, 22.6, 14.1, 14.1, 14.1. λ_max_ (CH_3_Cl/nm): 469 (log ε/dm^3^/mol/cm 4.47) HRMS (EI, *m*/*z*) anal. calcd. for C_86_H_110_N_4_O_6_S_2_+ (M^+^): 1359.7994 (100%); found: 1359.7940 (100%).

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
