# Peer review of "Pyrrolo[3,2-b]pyrrole-1,4-dione (IsoDPP) End Capped with Napthalimide or Phthalimide: Novel Small Molecular Acceptors for Organic Solar Cells"

_molecules, 2020, doi:10.3390/molecules25204700_

Round 1

Reviewer 1 Report

Pyrrolo[3,2-b]pyrrole-1,4-dione (IsoDPP) End Capped with Napthalimide or Phthalimide: Novel Small Molecular Acceptors for Organic Solar Cells

The authors reported two families of IsoDPPs functionalized with nafthalimide and phthalimide as acceptors in BHJ organic solar cells. All the compounds are well characterized, and the studies are well reported observing and effect in the photovoltaic parameters depending on the substitution in the IsoDPP substitution.

The paper is relevant among the organic solar cells readers and should be published in Molecules

Minor consideration:

DPV graphical data should be included to see the reduction values more accurately.

Eliminate the second decimal in the 13C-NMR data.

In the experimental section, include the UV-vis data with molar extinction coefficients.

Author Response

Reviewer 1:

The authors reported two families of IsoDPPs functionalized with napthalimide and phthalimide as acceptors in BHJ organic solar cells. All the compounds are well characterized, and the studies are well reported observing and effect in the photovoltaic parameters depending on the substitution in the IsoDPP substitution. The paper is relevant among the organic solar cells readers and should be published in Molecules.

Response: Many thanks to the reviewers for appreciation and valuable suggestion.

Minor consideration:

DPV graphical data should be included to see the reduction values more accurately.

Response: DPV and cyclic voltammetry provide data for calculating the E1/2. If the CV data is clear, then there is no need to for DPV measurements. From the current CV data, the oxidation and reduction onset values that are used to estimate the IP and EA are quite clear and no change is required.

Eliminate the second decimal in the 13C-NMR data.

Response:  The 13C-NMR data had been changed as requested - please see experimental section.

In the experimental section, include the UV-vis data with molar extinction coefficients.

Response: We have added UV-vis data with molar extinction coefficients values in the revised version - please see experimental section.

Reviewer 2 Report

The manuscript by Do et al. describes the synthesis, characterization and application of two novel compounds based on the IsoDPP core as acceptor materials for organic solar cells. Overall, the manuscript is well written and the experiments are clearly presented; the scientific work seems to have been conducted professionally and it is reasonably complete. For the above reasons I think this manuscript can be published in Molecules, but there are some issues that should be addressed before acceptance.

My main comments are:

1) The authors explicitly state in the introduction that it would be interesting to compare the IsoDPP-based compounds to their traditional DPP counterparts: yet, such comparison is not presented in the paper. Have compounds containing a traditional DPP core and the same (or very similar) capping groups been previously published? What were their physico-chemical properties, and what were the performances of the corresponding cells? I understand that comparison of solar cell efficiencies from one paper to another can be difficult due to the different fabrication and measurement conditions etc., but at least a comparison in terms of spectroscopic and electrochemical properties should be presented in the manuscript.

2) In the synthetic part, it seems to me that compound 5, although prepared via a known procedure, is not the same as that reported in reference 29. Therefore at least the last step of its preparation from known precursors should be presented in the experimental section, together with its characterization. In addition, although compounds 3 and 4 were not isolated, I think that at least the description of their 1H-NMR spectrum should be added in the experimental section, to provide at least some help to those researchers willing to repeat their preparation.

3) In Scheme 1 the exact reaction conditions and the reaction yields should be shown. In the experimental part compound numbers should be added in the procedures reported in paragraphs 4.3.1 and 4.3.2. In addition, the brute formulas of the identified compounds should be added in the description of the HRMS spectra.

4) Considering the final compounds, it can be seen that the smaller one (the "PI" compound) has a higher melting point compared to the larger one (the "NAI" compound), probably as an effect of tighter molecular packing. On the other hand, it also shows a smaller red-shift in the UV-Vis spectrum in going from solution to film, which could indicate a smaller variation of its geometry from the liquid to the solid state. Are these two data contradicting, or is there a common explanation for both?

5) The authors mention that their compounds in combination with P3HT give rise to cells with a Voc "ca. 350 mV higher" than that obtained with typical PCBM acceptors, as a result of their higher-lying LUMO. Yet, in several sources the LUMO of PCBM is reported at very similar levels (i. e. -3.9 eV vs. vacuum, see for example Acc. Chem. Res. 2012, 45, 723). So how is such a large difference possible? Moreover, the HOMO and LUMO values reported in the paper for P3HT are different from those sometimes found elsewhere (-4.7 and -2.7 eV vs. vacuum, for example, in the same reference cited above). Are these the consequence of different references? The author should include also the values for PCBM in the discussion and state clearly which reference they used to determine the reported values.

6) The EQE spectrum in the text and the AFM images in the supplementary materials are not discussed in the article. If such figures are presented, they should be described.

Author Response

Please see the file attached.

Reviewer 2:

The manuscript by Do et al. describes the synthesis, characterization, and application of two novel compounds based on the IsoDPP core as acceptor materials for organic solar cells. Overall, the manuscript is well written and the experiments are clearly presented; the scientific work seems to have been conducted professionally and it is reasonably complete. For the above reasons I think this manuscript can be published in Molecules, but there are some issues that should be addressed before acceptance.

Response: Many thanks to the reviewers for appreciation and valuable suggestion to improve the manuscript further.

My main comments are:

1) The authors explicitly state in the introduction that it would be interesting to compare the IsoDPP-based compounds to their traditional DPP counterparts: yet, such comparison is not presented in the paper. Have compounds containing a traditional DPP core and the same (or very similar) capping groups been previously published? What were their physico-chemical properties, and what were the performances of the corresponding cells? I understand that comparison of solar cell efficiencies from one paper to another can be difficult due to the different fabrication and measurement conditions etc., but at least a comparison in terms of spectroscopic and electrochemical properties should be presented in the manuscript.

Response: We have added a comparison between UV-Vis spectrum and electrochemical properties of IsoDPP-based compounds with respect to DPP counterparts in the optical and electrochemical properties sections as requested.

2) In the synthetic part, it seems to me that compound 5, although prepared via a known procedure, is not the same as that reported in reference 29. Therefore at least the last step of its preparation from known precursors should be presented in the experimental section, together with its characterization. In addition, although compounds 3 and 4 were not isolated, I think that at least the description of their 1H-NMR spectrum should be added in the experimental section, to provide at least some help to those researchers willing to repeat their preparation.

Response: The reviewer was correct with regard the reference, which was wrong. The complete synthetic procedure for compounds 5 is given below as per reported procedures (Macromolecules 2013, 46, 10, 3895–3906 and Macromolecules 2012, 45, 4511-4519).The scheme has been added in the supporting information for getting better clarity.  We have corrected the referencing and provided the synthesis and 1H and 13C NMR data for compound 5 in the supporting information.

3) In Scheme 1 the exact reaction conditions and the reaction yields should be shown. In the experimental part compound numbers should be added in the procedures reported in paragraphs 4.3.1 and 4.3.2. In addition, the brute formulas of the identified compounds should be added in the description of the HRMS spectra.

Response: We have added the data requested in the revised version.

4) Considering the final compounds, it can be seen that the smaller one (the "PI" compound) has a higher melting point compared to the larger one (the "NAI" compound), probably as an effect of tighter molecular packing. On the other hand, it also shows a smaller red-shift in the UV-Vis spectrum in going from solution to film, which could indicate a smaller variation of its geometry from the liquid to the solid state. Are these two data contradicting, or is there a common explanation for both?

Response: There is no contradiction in the data. The DSC measurements were performed on powder samples whereas, thin film absorption spectra were measured from thin films, which were spin-coated from solution onto glass substrates. Therefore, the morphology of the material in the different states can be different. No change to the manuscript is needed.

5) The authors mention that their compounds in combination with P3HT give rise to cells with a Voc "ca. 350 mV higher" than that obtained with typical PCBM acceptors, as a result of their higher-lying LUMO. Yet, in several sources the LUMO of PCBM is reported at very similar levels (i. e. -3.9 eV vs. vacuum, see for example Acc. Chem. Res. 2012, 45, 723). So how is such a large difference possible? Moreover, the HOMO and LUMO values reported in the paper for P3HT are different from those sometimes found elsewhere (-4.7 and -2.7 eV vs. vacuum, for example, in the same reference cited above). Are these the consequence of different references? The author should include also the values for PCBM in the discussion and state clearly which reference they used to determine the reported values.

Response: We thank the reviewer for raising this point. There are numerous values quoted for the IP and EA of the same material with the differences dependent on the measurement method. For example, some people use the onset to the oxidation/reduction processes (as we have in this paper) while others use the E1/2s. Conversion of the potentials from electrochemistry to IPs to EAs can also be done differently and hence comparisons can be fraught.

The published below review clearly highlights the open circuit voltage related challenges due to other influencing factors such as density of states or energetic disorder, charge transfer states, donor–acceptor interface, microstructure etc.1

  1. Naveen Kumar Elumalai and Ashraf Uddin, Energy Environ. Sci., 2016, 9, 391, ‘Open circuit voltage of organic solar cells: an in-depth review’

To make our statement clear, the text in the article is changed from  “It is notable that OSCs made with NAI-IsoDPP-NAI and PI-IsoDPP-PI acceptors show a Voc of ca. 350 mV higher with respect to those with PCBM acceptors, which could arise from their relatively smaller electron affinities.”

to

“It is notable that OSCs made with NAI-IsoDPP-NAI and PI-IsoDPP-PI acceptors show a Voc of ca. 350 mV higher with respect to those with PCBM acceptors37, despite similar LUMO levels.38 Such an increase in the observed values could then be attributed to other factors known to directly or indirectly influence the open circuit voltage in organic solar cells, such as density of states, energetic disorder, charge transfer states, donor–acceptor interface, microstructure etc.39

References

6) The EQE spectrum in the text and the AFM images in the supplementary materials are not discussed in the article. If such figures are presented, they should be described.

Response: We have added descriptions for EQE spectrum and AFM images in the revised version.

Reviewer 3 Report

This manuscript presents synthesis and photovoltaic applications of a new family of molecular acceptors. The molecular design based on IsoDPP is very impressive, because this structural unit found in natural sources represents a related analog of DPP and there is only a limited number of the IsoDPP materials explored for use in photovoltaics. The experimental results were analyzed precisely, and the main text was written properly to provide the molecular characteristics of the synthetic materials. The report shows only modest photovoltaic performances with relatively low PCEs and Jsc estimates. This point raises concerns about the morphological properties of the device active layers. Thus, the authors are encouraged to probe the relationship between the morphological features and photovoltaic properties for further discussion on the intense research efforts. The current manuscript should be appropriately revised to address the above issue before acceptance.

Author Response

Reviewer 3:

This manuscript presents synthesis and photovoltaic applications of a new family of molecular acceptors. The molecular design based on IsoDPP is very impressive, because this structural unit found in natural sources represents a related analog of DPP and there is only a limited number of the IsoDPP materials explored for use in photovoltaics. The experimental results were analyzed precisely, and the main text was written properly to provide the molecular characteristics of the synthetic materials.

  1. The report shows only modest photovoltaic performances with relatively low PCEs and Jsc estimates. This point raises concerns about the morphological properties of the device active layers. Thus, the authors are encouraged to probe the relationship between the morphological features and photovoltaic properties for further discussion on the intense research efforts. The current manuscript should be appropriately revised to address the above issue before acceptance.

Response: We have revised the morphology section of the paper and added some discussion around AFM data and device performance.

Round 2

Reviewer 2 Report

The authors responded to all the questions raised by this reviewer and made significant adjustments to the text. Therefore the manuscript can now be accepted for publication.

Reviewer 3 Report

The revised version of the manuscript provides sufficient improvement in experimental results to support the morphological and photovoltaic characteristics of the titled compounds. I therefore approve the acceptance of this paper without any further request.

This manuscript is a resubmission of an earlier submission. The following is a list of the peer review reports and author responses from that submission.